# EFRame: Deeper Reasoning via Exploration-Filter-Replay Reinforcement Learning Framework

## Abstract

Recent advances in reinforcement learning (RL) have significantly enhanced the reasoning capabilities of large language models (LLMs). Group Relative Policy Optimization (GRPO), a lightweight variant of Proximal Policy Optimization (PPO), improves efficiency but suffers from limited exploration and training instability, limiting its effectiveness on complex reasoning tasks. To address these challenges, we introduce EFRame, an Exploration-Filter-Replay framework that augments GRPO across three dimensions: additional rollouts enable deeper and more targeted exploration, online filtering removes low-quality samples to stabilize gradients and accelerate training, and experience replay amplifies rare yet informative trajectories for stable convergence. This unified framework establishes a principled training cycle that balances exploration, efficiency, and stability. Experiments on diverse reasoning benchmarks and evaluation settings demonstrate that EFRame achieves consistent gains, including a 37.9% improvement on Geometry3K over GRPO, and exceeds RL baselines under Pass@K settings. EFRame further supports fine-grained sample categorization and precise entropy control, highlighting it as a robust solution for advancing deeper reasoning in LLMs.

## 1 Introduction

Reinforcement learning (RL) has played a pivotal role in the post-training of large language models (LLMs) (GLM et al., 2024; Touvron et al., 2023). In the early stages, approaches such as Proximal Policy Optimization (PPO) and Direct Preference Optimization (DPO) were widely adopted under the Reinforcement Learning from Human Feedback (RLHF), effectively aligning model outputs with human preferences through preference-based rewards (Schulman et al., 2017; Rafailov et al., 2023; Zhong et al., 2024; Wang et al., 2024b). More recently, the focus has shifted towards Reinforcement Learning with Verifiable Rewards (RLVR), which seeks to provide more consistent and scalable supervision by leveraging structured, verifiable reward signals (Mroueh, 2025). Within this paradigm, DeepSeek-R1 introduced Group Relative Policy Optimization (GRPO), a lightweight and efficient variant of PPO that computes relative rewards within rollout groups to eliminate the need for the critic model (Shao et al., 2024; Liu et al., 2024; Guo et al., 2025). GRPO has demonstrated remarkable improvements in training efficiency and has shown promising potential in enhancing the reasoning capabilities of LLMs.

Despite its efficiency, GRPO suffers from limited exploration and training instability on complex reasoning tasks. We conducted extensive experiments and categorized the failed GRPO training cases into two types, as illustrated in Fig. 1, which shows representative runs on Qwen2.5-VL-7B-Instruct with the Geometry3K dataset. In the first example (GRPO-1 in Fig. 1), exploration was severely limited: the policy entropy continuously decreased from an initial value of 0.25 throughout training, making it difficult for the model to perform effective exploration and resulting in insufficient fine-tuning improvements. In the second example (GRPO-2 in Fig. 1), the training suffered an irreversible collapse, marked by an arch-shaped fall of rewards and accuracy during mid-training, followed by a breakdown of model performance that could not be recovered.

GRPO derives reward signals from relative advantages computed within rollout groups. This design inherently ties the training process to the accuracy of groups (Yue et al., 2025). When all sampled

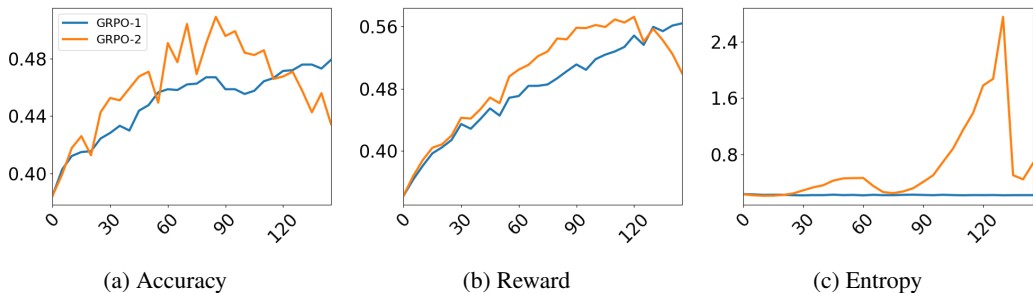

Figure 1: Two prominent issues of GRPO: limited exploration and training instability, when training on Qwen2.5-VL-7B-Instruct with the Geometry3K dataset. GRPO-1 and GRPO-2 share identical settings.

responses are either correct or incorrect, the relative advantage collapses to zero, yielding no effective learning signal. In long-horizon training, this advantage-vanishing effect amplifies gradient variance, which often manifests as gradient explosions and continuous reward degradation, eventually leading to irreversible training collapse (Yu et al., 2025; Li et al., 2025). Collectively, these issues constrain GRPO's ability to sustain exploration, preserve model accuracy, and maintain stability on challenging reasoning tasks.

To address the aforementioned issues, we propose **EFRame: an Exploration-Filter-Replay RL Framework**. For challenging prompts that cannot be effectively handled by regular rollouts, EFRame introduces an additional rollout to enable targeted exploration. During this process, the model generates a small number of high-quality positive samples along with a large number of low-quality negative samples. To reduce training variance and improve efficiency, we apply an online filtering mechanism to discard low-quality samples. Meanwhile, high-advantage samples are stored for experience replay to amplify their impact and enhance model stability.

The contributions of EFRame are threefold:

1. **Enhanced High-Difficulty Reasoning:** EFRame significantly improves the performance of GRPO on challenging reasoning tasks. By integrating exploration, filtering, and replay into a unified framework, it enables more complete and deeper reasoning processes in large language models.

2. **Sample-Type-Aware Training:** EFRame introduces a principled categorization of training samples to better understand their distinct roles in learning. It demonstrates that high-advantage, low-probability positive samples are crucial for triggering exploration and unlocking deeper knowledge, whereas low-advantage negative samples tend to drive premature convergence to suboptimal solutions and should be suppressed.

3. **Efficient Entropy Control:** EFRame provides a practical and effective method for entropy control by jointly tuning the additional rollout temperature and the number of replayed samples. Compared to traditional reward-based entropy regularization, this mechanism enables more flexible and stable entropy regulation, thereby facilitating both efficient exploration and stable convergence.

Overall, EFRame provides a systematic solution to the inherent trade-offs among stability, efficiency, and exploration capacity in RL. By effectively identifying and leveraging deeper latent knowledge, it enhances the expressiveness and learning potential of LLMs. Through the coordinated integration of targeted exploration, online filtering, and experience replay, EFRame not only addresses the limitations of GRPO but also establishes a principled and controllable training paradigm. Empirical results confirm its efficacy: on the challenging Geometry3K benchmark, EFRame outperforms GRPO by 37.9%, and achieves consistent improvements across diverse mathematical and multimodal reasoning benchmarks. Moreover, EFrame surpasses RL baselines (GRPO and DAPO) across all Pass@K settings, demonstrating strong exploration capabilities.

EFRame provides a systematic solution to the inherent trade-offs among training stability, efficiency, and exploration capacity in RL. By enabling the model to effectively identify and leverage deeper

latent knowledge, EFRame enhances the expressiveness and learning potential of large language models. Through the coordinated integration of targeted exploration, online sample filtering, and experience replay, the framework not only addresses the limitations of GRPO in handling complex reasoning tasks but also establishes a more principled and controllable training paradigm. Empirical evaluations confirm the efficacy of EFRame: on the challenging Geometry3K benchmark, it outperforms GRPO by 37.9%, and it achieves consistent gains across a wide range of mathematical and multimodal reasoning benchmarks, demonstrating strong generalizability and robustness.

## 2 RELATED WORK

The early application of RL in LLMs and MLLMs (OpenAI, 2023; Team et al., 2024; Zhu et al., 2023; Wei et al., 2023; Liu et al., 2023) was primarily reflected in RLHF, where algorithms such as PPO were employed to align language model outputs with human preferences and Direct Preference Optimization (DPO) (Rafailov et al., 2024) bypass the need for reward modeling and instead optimize the model policy directly from preference data. Recently, RL has advanced toward integrating rule-based reward mechanisms with sophisticated optimization techniques, as exemplified by models such as DeepSeek-R1 (Guo et al., 2025) and Kimi-1.5 (Team et al., 2025). As a lightweight variant of PPO, GRPO (Shao et al., 2024) significantly improves training efficiency, but at the cost of limited exploration and training instability.

A series of subsequent studies have introduced optimizations from GRPO. DAPO (Yu et al., 2025) and Dr.GRPO (Liu et al., 2025) adjust the policy gradient from a more global perspective, but they do not directly address the aforementioned issues. Entropy-driven exploration methods, e.g., (Cheng et al., 2025; Chen et al., 2025), introduce entropy regularization into reward or advantage functions to encourage exploration. However, this strategy only provides a coarse trend, making entropy difficult to control precisely and stably, while also introducing biases that are hard to estimate. Instead, we adopt an exploration method with limited bias—high quality sampling—which enables more effective exploration with controlled deviation. Online filtering accelerations, such as (Meng et al., 2025; Lin et al., 2025), further strengthen the efficiency advantage of GRPO, but their contribution to training effectiveness is almost negligible. Experience replay approaches, including VL-Rethinker (Wang et al., 2025), RLEP (Zhang et al., 2025), and RePO (Yan et al., 2025), enhance GRPO from the perspective of convergence stability. However, experience replay may also aggravate the entropy collapse problem in GRPO, and indiscriminate use can hinder the model's ability to explore more deeply. EFRame addresses these challenges by encouraging exploration through conditional sampling, accelerating training with online filtering, and achieving stable convergence via experience replay, thereby delivering consistent improvements to GRPO in exploration, stability, and convergence.

## 3 METHODOLOGY

To resolve the limited exploration and training instability in GRPO, we propose the EFRame, which consists of three main components: additional rollout, online filtering, and experience replay, as illustrated in Figure 2.

### 3.1 GRPO AND REGULAR ROLLOUT

Group Relative Policy Optimization (GRPO) directly estimates the advantage using group-normalized rewards. Given a question $q$ and a corresponding verifiable answer $a$, GRPO generates a set of responses $O = \{o_i\}_{i=1}^{G}$ using the policy model $\pi_{\theta_{old}}$, then optimizes the policy model by maximizing the following objective:

$$\mathcal{J}_{GRPO}(\theta) = \mathbb{E}_{(q,a)\sim\mathcal{D},\ O=\{o_i\}_{i=1}^{G_1}\sim\pi_{\theta_{old}}(\cdot|q)}$$

$$\frac{1}{G_1}\sum_{i=1}^{G_1}\frac{1}{|o_i|}\sum_{t=1}^{|o_i|}\left\{\min\left[r_{i,t}(\theta)\hat{A}_{i,t},\ \text{clip}(r_{i,t}(\theta), 1-\epsilon, 1+\epsilon)\hat{A}_{i,t}\right] - \beta D_{KL}[\pi_\theta||\pi_{ref}]\right\},$$

$$(1)$$

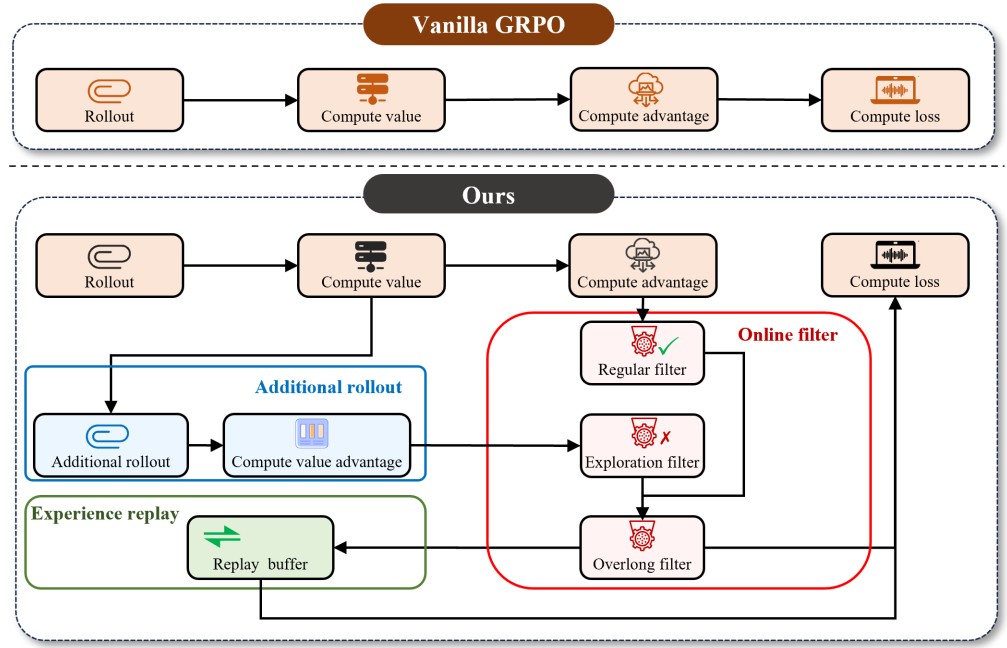

Figure 2: The overall workflow of EFRame builds upon GRPO by introducing three key components: additional rollout, online filter, and experience replay.

where $r_{i,t}(\theta) = \frac{\pi_\theta(o_{i,t}|q,o_i<t)}{\pi_{\theta_{old}}(o_{i,t}|q,o_i<t)}$ and $\hat{A}_{i,t} = \frac{R_i - \text{mean}(\{R_i\}_{i=1}^{G_1})}{\text{std}(\{R_i\}_{i=1}^{G_1})}$.

In EFRame, training begins with a regular rollout phase following this GRPO formulation. This initial rollout serves two key purposes:

- Obtain reward signals from regular (i.e., solvable) problems.

- Identify more challenging prompts that fail to yield positive samples, which will be addressed in subsequent stages.

## 3.2 Additional Rollout

GRPO computes loss over $G_1$ sampled outputs, making $G_1$ pivotal. Smaller $G_1$ limits learning capacity; while larger $G_1$ adds linear cost with limited benefit. For most questions, at least one positive sample can typically be found within a small number of rollouts. However, for more challenging problems, a larger number of samples and a higher temperature are often necessary to promote exploration and obtain meaningful reward signals.

To address this, we introduce an additional rollout mechanism: we perform an extra sampling round (e.g. $G_2$=20, temperature = 1.2) for challenging prompts.

**Definition 3.1.** In GRPO-based algorithms with binary rewards (Premise, omitted in the following), difficult prompts are defined as those for which all sampled responses within the initial $G_1$ rollouts ($G_1$ is a small integer, such as 5) fail to obtain a positive reward, where $\bar{\mathcal{D}} \subseteq \mathcal{D}$ and:

$$\bar{\mathcal{D}} = \left\{ (q,a) \; : \; | \{\{o_i\} \sim \pi_{\theta_{old}(\cdot|q)} \; : \; \text{is\_equivalent}(a, o_i)\}_{i=1}^{G_1} | = 0 \right\}.$$

The additional rollout is designed to activate deeper reasoning capabilities and increase the frequency of emergent "aha moments."

### 3.3 ONLINE FILTER

While additional rollout enhances the model's ability to explore complex prompts, it also introduces computational overhead and potential training noise. In addition to filtering overlong samples introduced in DAPO (Yu et al., 2025), we design two additional filtering strategies to further enhance the stability and efficiency of the training process.

**Regular Filter**: In the first-round rollout, when the prompt is of regular difficulty, we adopt a conservative filtering strategy that retains all regular samples $O_R$ to balance efficiency and performance.

**Definition 3.2.** Regular samples $O_R$ refer to any samples with a non-zero advantage on the prompt set $\mathcal{D}$:

$$O_R = \left\{ \{o_i\} \sim \pi_{\theta_{old}}(\cdot|q) : (q, a) \sim \mathcal{D},\ \hat{A}_{i,t} = \frac{R_i - \mathrm{mean}(\{R_i\}_{i=1}^{G_1})}{\mathrm{std}(\{R_i\}_{i=1}^{G_1})} \neq 0 \right\};$$

At this stage, due to the moderate difficulty of prompts, all regular samples—both positive and negative—provide informative learning signals that contribute meaningfully to training.

**Exploration Filter**: When the prompt becomes challenging, we encourage the model to explore extensively in order to discover correct answers. Excluding samples with zero advantage, the remains can be categorized into the following two types.

**Definition 3.3. High-quality samples** $O_H$ are positive samples for difficult prompts $\bar{\mathcal{D}}$:

$$O_H = \left\{ \{o_i\} \sim \pi_{\theta_{old}(\cdot|q)}^{t_a} : (q, a) \sim \bar{\mathcal{D}}, \hat{A}_{i,t} = \frac{R_i - \mathrm{mean}(\{R_i\}_{i=1}^{G_2})}{\mathrm{std}(\{R_i\}_{i=1}^{G_2})} > 0 \right\};$$

**low-quality samples** are negative samples for difficult prompts $\bar{\mathcal{D}}$:

$$O_L = \left\{ \{o_i\} \sim \pi_{\theta_{old}(\cdot|q)}^{t_a} : (q, a) \sim \bar{\mathcal{D}}, \hat{A}_{i,t} = \frac{R_i - \mathrm{mean}(\{R_i\}_{i=1}^{G_2})}{\mathrm{std}(\{R_i\}_{i=1}^{G_2})} < 0 \right\},$$

where $\pi_{\theta_{old}}^{t_a}$ denotes the distribution of $\pi_{\theta_{old}}$ under temperature $t_a$. In our setting, $t_a = 1.2$ by default.

In such an exploration-driven setting, low-quality samples do not provide meaningful reasoning signals. Instead, they tend to generate a large number of conflicting gradients that pull the policy toward spurious patterns useful only on the training distribution. Consequently, retaining these low-quality samples not only reduces optimization efficiency but also hampers generalization.

To summarize, the policy gradient under our framework can be expressed as:

$$\mathcal{J}(\theta) = \mathbb{E}\left[ \frac{1}{\sum\limits_{o_i \in O_R \cup O_H} |o_i|} \sum_{o_i \in O_R \cup O_H} \sum_{t=1}^{|o_i|} \left( \min\left\{ r_{i,t}(\theta)\, \hat{A}_{i,t},\ \mathrm{clip}\left( r_{i,t}(\theta), 1 - \epsilon_{\mathrm{low}}, 1 + \epsilon_{\mathrm{high}} \right) \hat{A}_{i,t} \right\} \right) \right].$$

$$(2)$$

To prevent token-length bias, we compute the policy loss using a token-mean aggregation strategy. Furthermore, following the conclusion of DAPO, and given that catastrophic forgetting is relatively minor in RL (Lai et al., 2025), we omit KL regularization during training to allow a more thorough optimization of the model.

### 3.4 EXPERIENCE REPLAY

While GRPO is fundamentally an online RL algorithm, we find that experience replay with high-quality offline samples remains highly effective. In particular, trajectories discovered during the second-round rollout—are often of higher quality due to deeper exploration and carry rich reward signals. By selectively storing high-quality samples in the replay buffer, we amplify their influence during training, particularly when their gradients are frequently clipped due to low sampling probabilities. Furthermore, by enabling the model to repeatedly leverage these rare yet informative samples, experience replay plays a pivotal role in enhancing the stability of entropy.

Table 1: Comparison of Pass@1 accuracy on text and multimodal reasoning benchmarks. EFRame consistently outperforms the RL baselines (GRPO and DAPO).

| Pass@1 | MATH | GSM8K | Minerva | Average |
|---|---|---|---|---|
| Qwen2.5-math-7B | 65.5 | 65.4 | 47.3 | 38.8 |
| GRPO | 77.6 | 87.1 | 29.0 | 64.57 |
| DAPO | 78.3 | 87.6 | 34.2 | 66.80 |
| **EFRame (ours)** | **80.4** | **87.9** | **35.7** | **68.00** |

| Pass@1 | MathVision | MathVerse | MathVista | We-Math | Average |
|---|---|---|---|---|---|
| Qwen2.5-VL-7B-Instruct | 24.87 | 43.83 | 66.30 | 62.87 | 49.47 |
| GRPO | **29.11** | 47.51 | 72.60 | 67.53 | 54.19 |
| DAPO | 27.92 | 48.48 | 72.30 | 69.08 | 54.45 |
| **EFRame (ours)** | 28.82 | **48.81** | **73.40** | **69.48** | **55.13** |

**Theorem 3.4** (Entropy Change under NPG Update). *If the actor policy $\pi_\theta$ is a tabular softmax policy updated via natural policy gradient (Kakade, 2001) with step size $\eta$. Then the change in policy entropy between two steps approximately satisfies:*

$$H(\pi_\theta^{k+1}) - H(\pi_\theta^k) \approx -\eta \cdot \mathbb{E}_{s \sim d_\mu^k} Cov_{a \sim \pi_\theta^k(\cdot|s)}\big[\log \pi_\theta^k(a \mid s), A^{\pi^k}(s,a)\big],$$

This theorem was first introduced by Liu (2025), and was organized and extended by Cui et al. (2025). Proof can be seen in Liu (2025) and Cui et al. (2025). $H$ indicates the policy entropy of policy model, and Cov denotes covariance, $\pi_\theta^k$ is the policy at step $k$, and $A^{\pi^k}(s,a)$ is the advantage function of action $a$ under state $s$. This result indicates that when high-advantage actions have low probability, they induce negative covariance, which increases entropy and reduces stability.

The high-quality samples identified through additional rollouts are typically responses to difficult prompts and thus are with relatively low probability. Using them only once yields limited and short-lived gradient effects, leading to an increase in model entropy. By incorporating these samples into the replay buffer, their probability of reoccurrence is effectively increased during training, enabling the model to transition from entropy increase to eventual convergence. In fact, the experience replay here serves to reduce policy entropy; detailed discussions can be found in the following subsection.

## 3.5 EXPERIMENTAL SETUP

**Training Datasets.** Our experiments are conducted on three datasets to evaluate the effectiveness of the algorithm across different domains and tasks: (1) **DAPO-17K**, a large-scale mathematics out-of-domain (ood) training dataset designed to evaluate the algorithm's performance on large language models (LLMs); (2) **Multimodal ood dataset** of size 6k randomly sampled from the dataset proposed in (Wei et al., 2025b;a), which is collected from established open-source resources, including Geometry3K (Lu et al., 2021a), GeoQA (Chen et al., 2021), GeoQA-Plus (Cao & Xiao, 2022), Geos (Seo et al., 2015), AI2D (Kembhavi et al., 2016), TQA (Kim et al., 2018), FigureQA (Kahou et al., 2017), TabMWP (Lu et al., 2022b), ChartQA (Masry et al., 2022), IconQA (Lu et al., 2021b), Clevr-Math (Lindström & Abraham, 2022), M3CoT (Chen et al., 2024), and ScienceQA (Lu et al., 2022a); (3) **Geometry3K**, focused on geometric problems (Lu et al., 2021a), which is used to evaluate the algorithm's in-domain performance and conduct ablation studies.

**Evaluation Benchmarks.** First, we evaluate the algorithm on four text-only reasoning benchmarks—AIME'24 (HuggingFaceH4, 2025), MATH (Lightman et al., 2023), GSM8K (Cobbe et al., 2021), and Minerva (Lewkowycz et al., 2022)—which are widely used to assess the problem-solving and reasoning capabilities of large language models (LLMs) across different levels of mathematical difficulty and domain coverage. These benchmarks span competition-level problems (AIME'24), advanced mathematics problem sets (MATH), grade-school word problems (GSM8K), and science/-mathematics questions from Minerva. Second, we evaluate on four open multimodal benchmarks: MathVision (Wang et al., 2024a), MathVista (Lu et al., 2023), MathVerse (Zhang et al., 2024), and We-Math (Qiao et al., 2024), which cover diverse problem types such as geometry, charts, and tables,

Table 2: Comparison of Pass@8, Pass@16, Pass@32, and Average@32 performance on AIME'24 and AIME'25. EFRame consistently outperforms the RL baselines across all evaluation settings.

| AIME'24 | Pass@8 | Pass@16 | Pass@32 | Average@32 |
|---|---|---|---|---|
| Qwen2.5-math-7B | 36.67 | 40.00 | 46.67 | 15.20 |
| GRPO | 50.83 | 53.33 | 60.00 | 27.81 |
| DAPO | 41.67 | 46.67 | 56.67 | 24.27 |
| **EFRame (ours)** | **54.17** | **58.33** | **66.67** | **30.52** |
| **AIME'25** | **Pass@8** | **Pass@16** | **Pass@32** | **Average@32** |
| Qwen2.5-math-7B | 19.17 | 20.00 | 23.33 | 7.39 |
| GRPO | 22.50 | 28.33 | 33.33 | 9.27 |
| DAPO | 26.67 | 33.33 | 36.67 | 8.54 |
| **EFRame (ours)** | **26.67** | **35.00** | **43.33** | **10.11** |

| Geometry3K | Qwen2.5-VL-7B-Instruct | EFRame | GRPO | DAPO |
|---|---|---|---|---|
| Accuracy | 38.44 | **55.41**(+16.97) | 50.75 (+12.31) | 49.09 (+10.65) |

(a) Accuracy      (b) Entropy      (c) Response length

Figure 3: On the Geometry3K dataset, our method not only achieves excellent performance but also demonstrates superior exploration capability and training stability.

spanning multiple subjects and levels of visual reasoning difficulty. Finally, we conduct a detailed analysis of the algorithms on the Geometry3K dataset to examine in-domain performance and perform ablation studies.

## 4 EXPERIMENTS

To validate the effectiveness of our methods, we present the experimental setup and results in the following sections. Our work is based on the EasyR1 and VeRL frameworks (Yaowei et al., 2025; Sheng et al., 2025), and we compare with the RL baselines GRPO (Shao et al., 2024) and its improved variant DAPO (Yu et al., 2025). See implementation details in Appendix A.

### 4.1 MAIN RESULTS

**Performance.** Table 1 shows that EFRame achieves consistent improvements across both text-only and multimodal reasoning benchmarks. Compared with GRPO and DAPO, EFRame demonstrates superior performance in both single-modal and multi-modal settings, highlighting its strong stability and generalization capability. On the in-domain Geometry3K dataset, the gain is even more pronounced: Fig. 3 reports a +16.97 improvement, corresponding to a 37.9% relative advantage over GRPO. As shown in Table 2, EFRame consistently surpasses both GRPO and DAPO under the Pass@8, Pass@16, and Pass@32 evaluation settings, highlighting its strong exploration capability.

**Exploration and Entropy Control.** As shown in Fig.3, our framework maintains batch-level entropy around 1, striking a balance between exploration and exploitation. In the later stages of training, when GRPO and DAPO tend to suffer from vanishing advantages, our method keeps the exploration

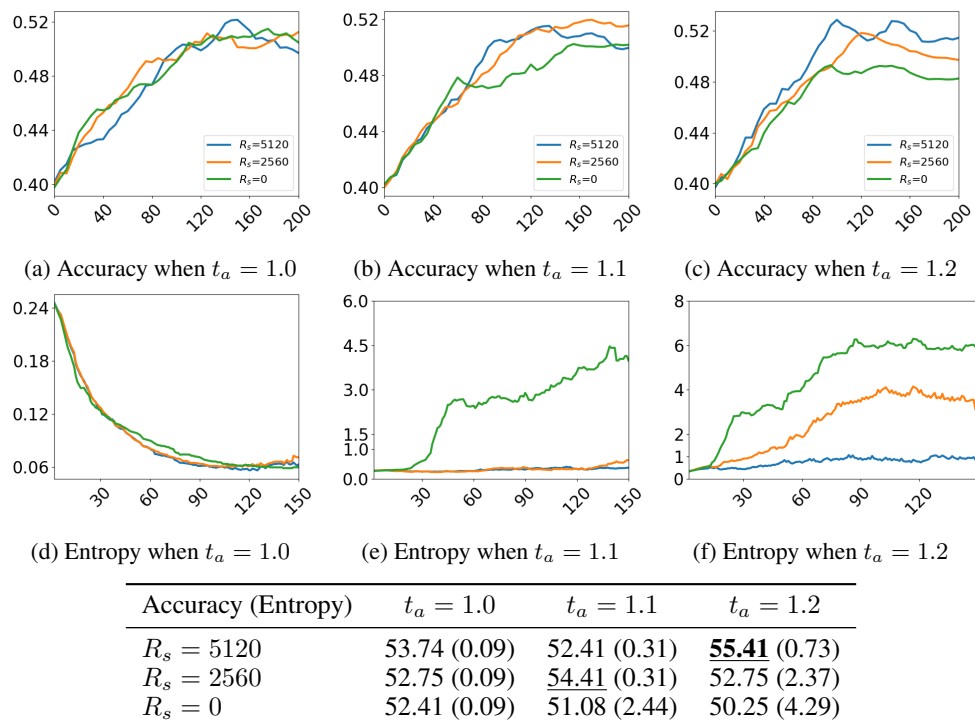

| Accuracy (Entropy) | $t_a = 1.0$ | $t_a = 1.1$ | $t_a = 1.2$ |
|---|---|---|---|
| $R_s = 5120$ | 53.74 (0.09) | 52.41 (0.31) | **55.41** (0.73) |
| $R_s = 2560$ | 52.75 (0.09) | 54.41 (0.31) | 52.75 (2.37) |
| $R_s = 0$ | 52.41 (0.09) | 51.08 (2.44) | 50.25 (4.29) |

Figure 4: Peak accuracy (first 200 steps) and average entropy (first 150 steps) on Geometry3K with Qwen2.5-VL-7B-Instruct under varying $t_a$ and $R_s$. Higher $t_a$ boosts exploration (entropy↑), larger $R_s$ aids convergence (entropy↓), and best performance arises from the balance state of entropy.

of informative reward signals. This is achieved by tuning two hyperparameters: the additional rollout temperature $t_a$ and the replay buffer size $R_s$. A higher $t_a$ broadens exploration by sampling low-probability yet high-reward trajectories, while a larger $R_s$ promotes convergence and suppresses excessive entropy growth. As shown in Fig.4, the two factors exert opposite influences on entropy dynamics, with optimal performance observed when entropy stabilizes at a moderate level.

*Claim* 1. **The combination of high-quality sampling and experience replay enables flexible and efficient entropy control.**

To further improve stability, the framework leverages additional rollouts to enrich reward signals, while an online filtering mechanism reduces training variance by removing low-quality samples, resulting in more robust gradients. Extensive experiments confirm that this design effectively avoids the collapse issues observed in GRPO and DAPO.

# 5 ABLATION STUDIES

To assess the contribution of each component in our framework, we conduct an ablation study to examine whether all three modules—additional rollout, online filtering, and experience replay—are essential for achieving the observed improvements in stability and overall performance.

## 5.1 ADDITIONAL ROLLOUT

Disabling the additional rollout module exposes a core trade-off in GRPO training. With few samples $G_1$, the reward signal is bounded by the capability of the base model, limiting its ability to solve difficult problems and suppressing "aha moments."

*Claim* 2. **High-quality samples can unlock the ability of deeper reasoning in RL.**

As shown in Fig. 5, training rewards plateau at low levels, constraining the accuracy. Early entropy remains low, reflecting limited exploration; once performance stalls, entropy rises but yields little

benefit, indicating difficulty escaping local optima. The additional rollout module addresses this by driving targeted deep exploration.

| Geometry3K | EFRame | w/o exploration | w/o filter | w/o replay |
|---|---|---|---|---|
| Accuracy | **55.41** | 50.41 (-5.00) | 51.91 (-3.50) | 49.58 (-5.93) |

(a) Accuracy        (b) Reward        (c) Entropy

Figure 5: The ablation study conducted on the Geometry3K dataset not only validates the effectiveness of our proposed method but also reveals the distinct roles played by samples of varying quality during the training process.

## 5.2 ONLINE FILTERING

Removing the online filtering module leads to unstable and inefficient training. Without filtering, all samples are treated equally: zero-advantage ones inflate gradient variance, and low-quality negatives distort the learning signal. This effect worsens in additional rollouts, where numerous incorrect samples overwhelm rare high-quality responses, making training slower, noisier, and prone to divergence.

*Claim* 3. **Low-quality samples from challenging prompts mislead optimization, driving premature convergence to suboptimal solutions and preventing further learning.**

As shown in Fig. 5, many negative samples initially boost training performance but provide little generalization gain, indicating faster convergence to local optima. Their forward and backward computation also adds overhead, lowering sample efficiency. Online filtering stabilizes training by preserving informative samples while discarding low-quality ones.

## 5.3 EXPERIENCE REPLAY

Disabling experience replay causes a sharp performance drop, even below the vanilla GRPO baseline (Fig. 5). This counterintuitive result shows that high-quality samples from additional rollouts can be harmful without a replay mechanism.

*Claim* 4. **Although high-quality samples drive high-entropy exploration, experience replay preserves their gradients, restoring stability and performance.**

High-quality samples usually have low generation probabilities: rewarding them shifts responses toward rare but correct outputs, raising entropy, and disturbing convergence. Their large advantage yet low probability also makes their gradients prone to clipping, leaving only fleeting signals that are quickly overwritten. Without replay, this transient effect amplifies uncertainty and destabilizes training. Experience replay ensures these rare samples repeatedly contribute effective gradients, guiding convergence along a stable trajectory.

## 6 CONCLUSION

We examined the limitations of GRPO on complex reasoning tasks and identified two major issues: limited exploration and training instability. To address these, we proposed EFRame, an Exploration-Filter-Replay framework that enriches exploration with additional rollouts, stabilizes training via online filtering, and improves convergence by replaying rare but informative samples. This cycle transforms RL from a "reinforcement lottery" into a more structured and controllable process.

Ablation studies show that high-advantage positives are crucial for deeper reasoning, while low-advantage negatives drive premature convergence, highlighting the value of selective filtering. EFRame further enables fine-grained entropy control by jointly tuning rollout temperature and replay ratios, avoiding biases of entropy regularization and mitigating collapse. Future work will focus on adaptive balancing of exploration and replay, more principled filtering, and integration with verifiable rewards and workflow-based training, paving a path toward more reliable and scalable reasoning in LLMs.

## DECLARATION OF AI

AI is only used for translation and language polishing in this paper.

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

# Appendix

## A  IMPLEMENTATION DETAILS

We initialize all policy models and conduct experiments on 8 A800 GPUs (40GB). Unless otherwise specified, we follow the default EasyR1 settings: maximum response length of 2048, global batch size 128, rollout batch size 512, rollout $G_1$=5, temperature 1.0, and learning rate $10^{-6}$. GRPO and our method adopt $\epsilon_{low} = \epsilon_{high} = 0.2$ and a binary reward, while DAPO includes $\epsilon_{high} = 0.3$ and an additional length-aware linear penalty (up to 0.5) for responses exceeding 1536 tokens. For our method, we perform additional rollouts with $G_2 = 20$, temperature = 1.2; responses with absolute advantage larger than 1 are stored in a replay buffer of size 5120, with experience replay triggered every 5 steps.

## B  LIMITATION

Although EFRame demonstrates substantial improvements across diverse benchmarks, it imposes a non-trivial requirement on the difficulty level of the training dataset. Specifically, EFRame benefits most when the dataset maintains a moderate difficulty level. If the dataset is overly challenging, both the regular rollout and the additional rollout stages suffer from excessively high failure rates, resulting in sparse reward signals. Conversely, if the dataset is too simple, the proportion of difficult prompts is low, rendering the additional rollout phase largely ineffective due to the scarcity of hard problems.

In this work, the selected datasets—DAPO-17K, the multimodal open-domain dataset, and Geometry3K—satisfy this moderate-difficulty criterion, enabling the additional rollout and filtering mechanisms to operate effectively. However, when training on the MATH-12K dataset, EFRame failed to deliver noticeable gains, with performance being nearly indistinguishable from the GRPO and DAPO baselines. This degradation is attributed to the dataset's excessive simplicity, which results in an insufficient number of high-quality samples discovered during additional rollout exploration. This observation underscores that the success of EFRame is contingent on the availability of a sufficient proportion of moderately difficult samples to balance exploration efficacy and training stability.

