# OpenReview forum: "EFRame: Deeper Reasoning via Exploration-Filter-Replay Reinforcement Learning Framework"
_ICLR.cc/2026/Conference — ICLR 2026 Conference Withdrawn Submission_

### Official Review · Reviewer_5FA8 · 2025-10-17

**Soundness:** 1
**Presentation:** 3
**Contribution:** 2
**Rating:** 2
**Confidence:** 4

**Summary:**

This paper proposes EFRame, a Exploration–Filter–Replay reinforcement learning framework designed to enhance the reasoning capability and stability of large language models (LLMs) during post-training. Building upon Group Relative Policy Optimization (GRPO), EFRame introduces three complementary modules: additional rollouts to promote targeted exploration for difficult prompts, online filtering to remove low-quality or zero-advantage samples (stabilizing gradients and improving efficiency), and experience replay to amplify the influence of rare but informative trajectories (mitigating entropy explosion and ensuring stable convergence).

**Strengths:**

- The paper is well written and easy to follow.
- The experiments are conducted on three diverse datasets and the gains are strong.
- The framework has three distinct parts which the authors conduct ablations by isolating the effect of each component.

**Weaknesses:**

- While well-engineered, the framework primarily combines known components (resampling, filtering, replay buffer) on top of the existing GRPO framework rather than introducing a fundamentally new optimization principle.
- The paper lacks theoretical justifications, and some claims are poorly supported:
    - In lines 243 - 248, "low-quality samples are significantly more numerous than high-quality ones, ... the informative signal from high-quality samples may be drowned out by chaotic updates from low-quality ones". I am not sure if it is correct. Although there are more low-quality samples, the absolute value of their advantages is much closer to 0 than the high-quality ones. In other words, we assign larger weights to the high-quality ones when updating than to the low-quality ones. As you stated in Theorem 3.4, the sum of their advantages should be 0, which means that we put the same amount of "weight" onto low and high quality responses.
    - Theorem 3.5 uses a theorem under the tabular setting with NPG. I am not sure if it can be directly applied here.
- The experiment settings are a bit "toy" and not really realistic. The paper uses Qwen 2.5, which is a bit old, a relatively short context of 2k tokens, and the number of rollouts ($G_1$) is 5, while DAPO and other recent recipes set it to 16. The scale of the experiments is not large enough to showcase the effectiveness of the method.
- Inconsistent experiment results.
    - Could the authors clarify how evaluation is performed on AIME and MATH? Is it based on pass@1 only? The AIME results suggest this may be the case, since the reported numbers are multiples of 1/30 (AIME has 30 questions). If so, the variance of pass@1 is quite high, and it would be more robust to report pass@32 or a similar metric.
    - For MATH, if the metric is indeed pass@1, it is unclear how results such as 65.5 and 78.3 were obtained. The test set contains 500 questions, so the accuracy should be a multiple of 1/500, and fractional correctness (e.g., half a question) is not meaningful. You cannot answer a question half correctly.
    - From Figure 5, we can see that, at step 100, the reward of the orange line is much higher than the blue line, while the accuracy is the other way around. I wonder if it is also due to the large variance of the evaluation.
    - DAPO takes more time for each step as it keeps resampling. However, under the same number of steps, the performance of DAPO should be higher than GRPO since, for each batch, it contains more gradient signals as zero-advantage prompts are all filtered out. Could the authors provide an explanation for why DAPO performs worse than GRPO in their experiments?
- Small typos:
    -  Figure 3 (b) and (c), EFR should be EFRame.

**Questions:**

Please refer to the weaknesses.

---

> ### Author Response · Authors · 2025-11-17
>
> ### Dear Reviewer 5FA8
>
> We sincerely thank the reviewer for the detailed and constructive comments. We have uploaded a revised version of our paper. All updated contents are marked in blue for easy reference.
>
> ### Response to W1
>
> Existing works typically use only one of the components we study and do not analyze the conditions under which each component is effective. Our contribution is proposing a complete and coherent framework, rather than a single isolated technique. By integrating these components, the framework jointly improves performance and stability and produces synergistic—rather than merely additive—effects (3 − 1 = 0 or even < 0).
>
> - EFRame enables principled control over policy entropy, allowing the algorithm to adjust exploration levels in a structured manner, which was not easy to implement before.
> - Experience replay consistently reduces policy entropy (Fig. 4), a useful but previously unreported observation.
> - Experience replay becomes more effective specifically under high-entropy conditions, another meaningful finding not discussed in prior work.
> - Moderate filtering leads to performance improvements (Fig. 5), which has not been systematically analyzed before.
> - High-entropy exploration without experience replay can harm policy stability (Fig. 5).
>
> These observations have not been discussed in prior work and form an important contribution of our study.
>
> ### Response to W2
>
> -We agree that the original phrasing was not sufficiently precise, and we have revised that part accordingly.
>
> -The tabular NPG theorem we provide currently represents the only available theoretical analysis for this setting. Extending theory to LLMs or VLMs is substantially more difficult, and our tabular result serves as a meaningful qualitative foundation.
>
> ### Response to W3
>
> The effect of sequence length is minimal: the response-clip ratio remains below 0.05 (typically \\\\< 0.02) throughout training.
>
> The choice of G = 5 follows the default settings of EasyR1 and VERL. Our results show that GRPO and DAPO achieve performance comparable to previous work, confirming that our settings do not artificially suppress their performance or impair their optimization capability. EFRame’s choices of G1=5 and G2=20 are also motivated by efficiency considerations. Using an overall group size of 16 would incur excessively high computational costs. This demonstrates that EFRame is a framework that balances multiple factors rather than optimizing a single metric in isolation.
>
> ### Response to W4
>
> -We have now added pass@k metrics (including pass@32) on AIME24 and AIME25 in the revised version.
>
> -The reviewer refers to MATH500, which is a subset of the full MATH (5k) dataset; our use of the larger dataset is standard.
>
> -The observed phenomenon is not caused by high variance: A more intuitive explanation is that negative samples on difficult problems help maximize reward on the training set but offer no benefit to generalization, causing early convergence to a local optimum and degraded test performance.
>
> -DAPO uses clip-higher, which increases variance and may hurt stability.  As a result, its performance on some tasks can occasionally fall behind GRPO. Similar behavior is also observed in Table 3 of [1].
>
> We sincerely appreciate the reviewer’s careful evaluation and constructive feedback.
>
> [1] Li G., Lin M., Galanti T., DisCO: Reinforcing Large Reasoning Models with Discriminative Constrained Optimization, arXiv:2505.12366 (2025).

---

> > ### Comment · Reviewer_5FA8 · 2025-11-19
> >
> > Thank you for providing the additional analysis and clarifications. However, I still find that the experimental scale is not sufficient to convincingly demonstrate the effectiveness of the proposed method, and the motivation behind the approach remains insufficiently supported from both theoretical and empirical perspectives. As a result, I will maintain my current score.

---

### Official Review · Reviewer_PtLK · 2025-10-30

**Soundness:** 3
**Presentation:** 2
**Contribution:** 1
**Rating:** 2
**Confidence:** 4

**Summary:**

This paper proposes EFRame, a framework that enhances the reasoning ability of LLMs through an exploration–filter–replay mechanism built on top of GRPO. The idea is to generate more diverse samples, filter low-quality responses online, and replay high-quality trajectories to improve stability and exploration efficiency. Experiments are conducted on Qwen models across several reasoning benchmarks.

**Strengths:**

1. The proposed Exploration–Filter–Replay framework is conceptually clear and easy to follow.

2. The method improves training stability and reasoning accuracy compared to GRPO baselines.

3. The ablation experiments provide useful insight into the contribution of each component.

**Weaknesses:**

1. Limited novelty: Similar mechanisms have already been explored in RLEP [1], RePO [2] and VL-Rethinker [3], which all employ replay-based or filtering strategies to stabilize reinforcement learning for reasoning tasks.

2. Baseline insufficiency: The paper does not compare against these closely related works [1–3], making it unclear how much gain is attributable to EFRame itself.

3. Lack of exploration metrics: The claimed improvement in exploration is not supported by pass@k, a standard evaluation metric for reasoning diversity.

4. Model limitation: Experiments are restricted to two Qwen models, with no tests on other LLM families (e.g., Llama, Deepseek).

5. Benchmark limitation: The paper omits newer reasoning benchmarks such as AIME25, MMT-Feb24, HMMT-Feb25, and CMIMC25.

6. Data contamination risk: Recent research shows that Qwen2.5 is susceptible to data leakage on certain reasoning benchmarks, raising doubts about evaluation improvements [4].

7. Lack of theoretical explanation: The paper provides no deeper analysis of why the combined exploration–filter–replay design leads to consistent improvement beyond empirical evidence.

**Questions:**

1. How does EFRame differ algorithmically from RLEP [1] , RePO [2] and VL-Rethinker [3]

2. How about the results on pass@k?

3. Have the authors tested EFRame on other model families to confirm generality?

4. Would the conclusions hold on newer benchmarks such as AIME25 or HMMT-Feb25?



[1] Zhang et al., RLEP: Reinforcement Learning with Experience Replay for LLM Reasoning, arXiv:2507.07451

[2] Li et al., RePO: Replay-Enhanced Policy Optimization, arXiv:2506.09340

[3] Wang et al., VL-Rethinker: Incentivizing Self-Reflection of Vision-Language Models with Reinforcement Learning, arXiv:2504.08837

[4] Wu et al., Reasoning or Memorization? Unreliable Results of Reinforcement Learning Due to Data Contamination, arXiv:2507.10532

---

> ### Author Response · Authors · 2025-11-17
>
> ### Dear Reviewer PtLK
> ﻿We sincerely thank the reviewer for the detailed and constructive comments. We have uploaded a revised version of our paper. All updated contents are marked in blue for easy reference. ﻿ Our responses are as follows. ﻿
> ### Response to W1 & W2
> We acknowledge that adaptive sampling, online filtering, and experience replay have appeared in prior literature. However, the works cited by the reviewers are contemporaneous papers (recent arXiv and NeurIPS 2025), which—per the ICLR 2026 Reviewer Guide—should not be required as baselines or comparison. Our contribution is proposing a complete and coherent framework, rather than a single isolated technique. By integrating these components, the framework jointly improves performance and stability and produces synergistic—rather than merely additive—effects (3 − 1 = 0 or even < 0).
> ﻿
> - EFRame enables principled control over policy entropy, allowing the algorithm to adjust exploration levels in a structured manner, which was not easy to implement before.
> ﻿
> - Experience replay consistently reduces policy entropy (Fig. 4), a useful but previously unreported observation.
> ﻿
> - Experience replay becomes more effective specifically under high-entropy conditions, another meaningful finding not discussed in prior work.
> ﻿
> - Moderate filtering leads to performance improvements (Fig. 5), which has not been systematically analyzed before.
> ﻿
> - High-entropy exploration without experience replay can harm policy stability (Fig. 5).
> ﻿
> ﻿
> ### Response to W3
> ﻿
> The reviewer’s suggestion regarding pass@k evaluation is reasonable. We have now added pass@k metrics (including pass@32) on AIME24 and AIME25 in the revised version
> ﻿
> ###Response to W4
> ﻿
> We conduct experiments on both an LLM (Qwen2.5-Math-7B) and a VLM (Qwen2.5-VL-7B-Instruct) to demonstrate the generality of our method across architectures and modalities.
> ﻿
> ### Response to W5
> We would like to emphasize that our current evaluation already covers the most widely-recognized datasets:
> ﻿
> - Text reasoning: AIME25, AIME24, MATH, GSM8K, Minerva
> - Multimodal reasoning: GEO3K, MathVision, MathVista, MathVerse, We-Math
> ﻿
> This breadth is sufficient to validate the generalization of our method. Many published works, such as [1], [2], and [3], include only the four datasets MathVision, MathVista, MathVerse, and We-Math.
> ﻿
> ### Response to W6
> ﻿
> 1. Qwen2.5-VL-7B-Instruct is not contaminated, and most multimodal experiments rely on it.
> 2. The contamination claim in [4] is controversial and not peer-reviewed; it mainly explains why random reward might improve performance, which is unrelated to our setting.
> ﻿
> ### Response to W7
> ﻿
> Theoretical analysis on LLMs or even VLMs is extremely challenging. Therefore, our tabular NPG analysis serves as a qualitative theoretical foundation. Our empirical results across multiple benchmarks further validate the conclusions drawn from this theoretical setting.
> ﻿
> [1] Wei L, Li Y, Wang C, et al. First SFT, Second RL, Third UPT: Continual Improving Multi-Modal LLM Reasoning via Unsupervised Post-Training[C]//The Thirty-ninth Annual Conference on Neural Information Processing Systems.
> ﻿
> [2] Liu X, Ni J, Wu Z, et al. Noisyrollout: Reinforcing visual reasoning with data augmentation[J]. arXiv preprint arXiv:2504.13055, 2025. (NIPS2025)
> ﻿
> [3] Shen Z, Yu Q, Li J, et al. EvolvedGRPO: Unlocking Reasoning in LVLMs via Progressive Instruction Evolution[C]//The Thirty-ninth Annual Conference on Neural Information Processing Systems.

---

> > ### Comment · Reviewer_PtLK · 2025-11-17
> >
> > Thanks for your reply, but it didn't solve my main concerns, so I decided to keep my score.

---

### Official Review · Reviewer_KSq2 · 2025-10-30

**Soundness:** 3
**Presentation:** 3
**Contribution:** 2
**Rating:** 4
**Confidence:** 4

**Summary:**

Vanilla Group Relative Policy Optimization (GRPO) suffers from limited exploration and training instability. To address these, the authors introduce EFRame that augments GRPO via three components: (1) additional rollouts to promote exploration, (2) online filtering removes low-quality samples to stabilize gradients, and (3) experience replay for stable convergence. Through these mechanisms,  EFRame balances exploration, efficiency, and stability, finally achieving 4.6% improvement on Geometry3K over vanilla GRPO.

**Strengths:**

1. Authors provide recipe for stable RL training which includes additional rollouts with higher temperature, online filtering, and experience replay. I believe it's a promising research direction.

2. This paper provides detailed analysis of each introduced mechanism based on the current challenges of GRPO, which is well motivated and reasonable.

3. This paper is well organized and easy to follow.

**Weaknesses:**

I discuss the weaknesses of originality and experiments. Weaknesses marked with **W** are key concerns that might affect the final rating, while weaknesses marked with **M** may have minor impact on my rating.

### Originality
**[M1]** The core ideas used in this work, *i.e.*, adaptive sampling for hard problems [1][2], online filter [2][3] and experience relay [4][5], have been explored in prior literature. This work combines these existing ideas well, but it's not very inspiring to me.

### Experiment
**[W1] Limited evaluation domains.** I notice that the training dataset includes math domain (DAPO-17k) and other general domains (*e.g.*, ScienceQA, ChartQA). However, the evaluation only focuses on mathematical reasoning domains, which raises concerns about its generalization ability. I suggest adding general reasoning tasks like MMLU-STEM, MMLU-Pro, GPQA, MMMU, and DocVQA to further validate the effectiveness of the proposed method.

**[W2] Limited model backbones.** This paper only use Qwen2.5-7B series (*i.e.*, Qwen2.5-math-7b, Qwen2.5-VL-7B-Instruct) to conduct the experiments. However, recent studies reveal that there may be potential data contamination in the Qwen model [6]. Consequently, conclusions derived from contaminated benchmarks (MATH-500, GSM8K) on Qwen2.5 series may be unreliable. The transferability of the proposed method to different models like Llama or Gemma warrants a more in-depth investigation.

**[M2] Concerns on Performance Gains.** I notice that EFRame improves ~1.0% over baselines on most benchmarks (Table 1). Does this suggest a possible limit to the power of this method to discover new reasoning patterns?

**[W3] Concerns on Experimental Settings.** I notice the maximum response length is set to 2,048 in RL training (Appendix A). But in DAPO, the default maximum response length is 20,480. So I wonder if the settings unintentionally impaired the exploration capabilities of baselines like GRPO and DAPO.

**[M3] Missing hyperparameters.** I don't find any information about the hyperparameters for the evaluation. What's the maximum response length, temperature, and sample numbers in evaluation? Besides, I don't see the prompt template used in training.

**[W4] Missing computational costs discussion.** In the vanilla GRPO baseline, what is the number of responses sampled for each question? It seems that the introduction of additional rollout and experience replay may bring more computational overhead. I suggest reporting relevant computational costs clearly.

---

[1] Optimizing Chain-of-Thought Reasoners via Gradient Variance Minimization in Rejection Sampling and RL. NeurIPS, 2025

[2] DAPO: An Open-Source LLM Reinforcement Learning System at Scale. arXiv preprint arXiv:2503.14476

[3] MM-Eureka: Exploring Visual Aha Moment with Rule-based Large-scale Reinforcement Learning. arXiv preprint arXiv:2503.07365

[4] RLEP: Reinforcement Learning with Experience Replay for LLM Reasoning. arXiv preprint arXiv:2507.07451

[5] Learning to reason under off-policy guidance. arXiv preprint arXiv:2504.14945

[6] Reasoning or Memorization? Unreliable Results of Reinforcement Learning Due to Data Contamination. arXiv preprint arXiv:2507.10532

**Questions:**

1. What's the setting of GRPO-1 and GRPO-2 in the Introduction? Are there any differences?

2. How many numbers of old responses are used to replay?  Can authors provide more details on the process of replay?

3. How will EFRame handle samples without positive signals after additional rollout?

---

> ### Author Response · Authors · 2025-11-14
>
> ### Dear Reviewer KSq2
> ﻿
> ﻿
> We sincerely thank the reviewer for the detailed and constructive comments. We have uploaded a revised version of our paper. All updated contents are marked in blue for easy reference.
> ﻿
> Our responses are as follows.
> ﻿
> ### M1. Originality of Core Modules
> ﻿
> We acknowledge that adaptive sampling, online filtering, and experience replay have appeared in prior literature. However, the works cited by the reviewers are contemporaneous papers (recent arXiv and NeurIPS 2025), which—per the ICLR 2026 Reviewer Guide—should not be required as baselines or comparison. Our contribution is proposing a complete and coherent framework, rather than a single isolated technique. By integrating these components, the framework jointly improves performance and stability and produces synergistic—rather than merely additive—effects (3 − 1 = 0 or even < 0).
> ﻿
> - EFRame enables principled control over policy entropy, allowing the algorithm to adjust exploration levels in a structured manner, which was not easy to implement before.
> ﻿
> - Experience replay consistently reduces policy entropy (Fig. 4), a useful but previously unreported observation.
> ﻿
> - Experience replay becomes more effective specifically under high-entropy conditions, another meaningful finding not discussed in prior work.
> ﻿
> - Moderate filtering leads to performance improvements (Fig. 5), which has not been systematically analyzed before.
> ﻿
> - High-entropy exploration without experience replay can harm policy stability (Fig. 5).
> ﻿
> ### W1. Limited Evaluation Domains
> ﻿
> Our experiments already cover nine test sets across text-only and multimodal domains.
> ﻿
> We further evaluate on four representative multimodal reasoning benchmarks:
> ﻿
> - MathVision — 3,040 visual competition problems across 12 grades
> - MathVista — 1,000 geometry, chart, and table questions
> - MathVerse — 3,940 diagram-based geometry problems
> - We-Math — 1,740 categorized visual problems across 67 knowledge concepts
> ﻿
> These datasets span diverse modalities and difficulty levels, providing a broad and rigorous evaluation. Many published works, such as [1], [2], and [3], include only the four datasets MathVision, MathVista, MathVerse, and We-Math.
> ﻿
> ﻿
> We believe this coverage is sufficient, and additional benchmarks mentioned by the reviewer can be added if reviewer deemed necessary.
> ﻿
> ### W2. Limited Model Backbones
> ﻿
> First, we confirm that Qwen2.5-VL-7B-Instruct is free from contamination and is used in most multimodal experiments.
> ﻿
> The cited preprint [6] is not peer-reviewed, and recent works still widely adopt Qwen2.5-Math-7B, suggesting that the contamination claim is far from being established. After carefully reading [6], we note that its abstract explicitly states: “Surprisingly, some studies even suggest that random or incorrect reward signals can enhance performance.” The purpose of [6] is to criticize the unusual performance gains observed under unsupervised or reward-free settings.
> ﻿
> However, our setting is fully supervised, and under fair comparison we consistently outperform GRPO and DAPO. Therefore,  we believe our conclusions remain reliable.
> ﻿
> ### M2. Performance Gains
> ﻿
> Performance variation across benchmarks is expected, but our gains are consistent and clear, particularly on AIME’24, MATH, Minerva, and Geometry3K, showing robustness.
> ﻿
> ### W3. Experimental Settings
> ﻿
> The response-clip ratio is always below 0.05 (typically < 0.02), indicating negligible impact on training stability.
> ﻿
> ### M3. Missing Hyperparameters
> ﻿
> All evaluations use the following configuration:
> ﻿
> - Max length: 2048
> - Temperature: 0.3
> - Template: EasyR1 default
> ﻿
> ### W4. Computational Costs
> ﻿
> Our method runs at 1.35× GRPO time per step, a reasonable overhead for replay and filtering.
> ﻿
> ### Q1.
> ﻿
> Both trajectories come from exactly the same training setup, which indicates that GRPO may suffer from two issues simultaneously: instability and insufficient exploration.
> ﻿
> ### Q2.
> ﻿
> After every 5 steps (sampling 2560 × 5 responses), we replay 5120 samples via a FIFO queue, ensuring that recently collected trajectories are reused while older ones are gradually discarded. This design maintains freshness in the replay buffer and prevents the policy from being influenced by outdated or overly off-policy samples.
> ﻿
> ### Q3.
> ﻿
> Samples without positive signals are discarded, as they are too difficult and off-policy, offering little learning benefit while risking distribution shift. Such cases are rare.
> ﻿
> We sincerely appreciate the reviewer’s careful evaluation and constructive feedback.
> ﻿
> [1] Third UPT: Continual Improving Multi-Modal LLM Reasoning via Unsupervised Post-Training, NIPS 2025.
> ﻿
> [2] Noisyrollout: Reinforcing visual reasoning with data augmentation. NIPS2025.
> ﻿
> [3] EvolvedGRPO: Unlocking Reasoning in LVLMs via Progressive Instruction Evolution. NIPS2025
> ﻿
> [6] Reasoning or Memorization? Unreliable Results of Reinforcement Learning Due to Data Contamination. arXiv preprint arXiv:2507.10532

---

### Official Review · Reviewer_3JzJ · 2025-11-01

**Soundness:** 2
**Presentation:** 3
**Contribution:** 2
**Rating:** 2
**Confidence:** 4

**Summary:**

The paper propose a framework of exploration and filter sample in RLVR. The experimental results show that it can enhance the performance of RL reasoning.

**Strengths:**

- This article focuses on an important issue, the significance of exploration for RL.
- The idea is very simple and extensible to prior methods.

**Weaknesses:**

- The experimental design is relatively weak, with too few baselines — aside from the fundamental algorithm GRPO, the comparison includes only one method of the same type (DAPO). The experimental analysis is also insufficient.
- The paper does not verify scalability, such as testing across different model architectures or sizes.
- All chosen benchmarks are standard math tasks, without any out-of-distribution (OOD) tasks to demonstrate the effectiveness of exploration.

**Questions:**

As noted in the weaknesses, the paper need to add more experiments and analyses:
- Provide clearer differentiation from concurrent work with detailed comparisons.
- Expand experiments to include more baselines, models, and diverse domains.
- Conduct more thorough case analysis.

If the authors can address  the above concerns in a revision, I would be willing to reconsider my assessment.

---

> ### Author Response · Authors · 2025-11-14
>
> ### Dear Reviewer 3JzJ
>
> ﻿We sincerely thank the reviewer for the detailed and constructive comments. We have uploaded a revised version of our paper. All updated content, figures, and tables are marked in blue for easy reference.
>
> Our responses are as follows.
>
> ### Response to Weakness 1
> ﻿
> ﻿
> GRPO and DAPO are the two most widely used RL baselines for LLMs.  Other methods such as RLEP or RePo, as concurrent works of our method that include only one of our three components, are also not peer-reviewed and lack stable public implementations. According to the ICLR 2026 Guide, authors are not expected to compare against work released within the last 2 months. Therefore, they are discussed in Related Work but not used as baselines.
> We acknowledge that adaptive sampling, online filtering, and experience replay have been explored before. Our contribution lies in integrating them into a unified framework that jointly optimizes performance and stability.The effect is synergistic rather than additive (3 − 1 = 0 or even < 0).
> ﻿
> - Experience replay consistently reduces policy entropy (Fig. 4), a useful but previously unreported observation.
> ﻿
> - Experience replay becomes more effective specifically under high-entropy conditions, another meaningful finding not discussed in prior work.
> ﻿
> - Moderate filtering leads to performance improvements (Fig. 5), which has not been systematically analyzed before.
> ﻿
> - High-entropy exploration without experience replay can harm policy stability (Fig. 5), a phenomenon that, to our knowledge, has not been reported in previous studies.
> ﻿
> ### Response to Weakness 2
> ﻿
> Our experiments already cover different architectures — Qwen2.5-Math-7B (LLM) and Qwen2.5-VL-7B-Instruct (VLM). These models differ greatly in architecture and modality, so achieving consistent improvements on both verifies scalability.
> ﻿
> ### Response to Weakness 3
> ﻿
> Our benchmarks include both math reasoning and multimodal geometry problems, which evaluate reasoning and optimization ability without external tools. As stated in lines 317–318, the training set is out-of-domain relative to the evaluation tasks, sufficiently demonstrating generalization.
> ﻿
> ### Response to Question 1
> ﻿
> The comment about insufficient analysis is too general. We kindly ask the reviewer to specify which part lacks analysis so we can strengthen it accordingly.
> ﻿
> ### Response to Question 2
> ﻿
> We agree that non–peer-reviewed methods are not suitable for direct comparison. The only exception, VL-ReThinker, includes an SFT stage, making direct comparison unfair. The replay-only variant has already been covered in our ablation study (“w/o additional rollout”).
> ﻿
> ### Response to Question 3
> ﻿
> Our work already covers both LLM and VLM, text and multimodal tasks, 3 training sets, and 9 testing sets. The benchmarks we use are far more comprehensive than those in several papers published at NeurIPS 2025[1,2,3]. If the reviewer believes this scope is insufficient, we would appreciate more concrete guidance on what additional scale would be considered adequate.
>
> We sincerely appreciate the reviewer’s careful evaluation and constructive feedback.
>
> ### Reference﻿
>
> [1] Wei L, Li Y, Wang C, et al. First SFT, Second RL, Third UPT: Continual Improving Multi-Modal LLM Reasoning via Unsupervised Post-Training[C]//The Thirty-ninth Annual Conference on Neural Information Processing Systems.
> ﻿
> [2] Liu X, Ni J, Wu Z, et al. Noisyrollout: Reinforcing visual reasoning with data augmentation[J]. arXiv preprint arXiv:2504.13055, 2025. (NIPS2025)
> ﻿
> [3] Shen Z, Yu Q, Li J, et al. EvolvedGRPO: Unlocking Reasoning in LVLMs via Progressive Instruction Evolution[C]//The Thirty-ninth Annual Conference on Neural Information Processing Systems.

---

> > ### Comment · Reviewer_3JzJ · 2025-11-17
> >
> > Thanks for your reply, but it didn't solve my main concerns, so I decided to keep my score. Please use more comprehensive experimental results to answer my questions.

---

### Note · Authors · 2025-12-23

I have read and agree with the venue's withdrawal policy on behalf of myself and my co-authors.